# Molecular Characterization Reveals Subclasses of 1q Gain in Intermediate Risk Wilms Tumors

**DOI:** 10.3390/cancers14194872

**Published:** 2022-10-05

**Authors:** Ianthe A. E. M. van Belzen, Marc van Tuil, Shashi Badloe, Eric Strengman, Alex Janse, Eugène T. P. Verwiel, Douwe F. M. van der Leest, Sam de Vos, John Baker-Hernandez, Alissa Groenendijk, Ronald de Krijger, Hindrik H. D. Kerstens, Jarno Drost, Marry M. van den Heuvel-Eibrink, Bastiaan B. J. Tops, Frank C. P. Holstege, Patrick Kemmeren, Jayne Y. Hehir-Kwa

**Affiliations:** 1Princess Máxima Center for Pediatric Oncology, 3584 CS Utrecht, The Netherlands; 2Oncode Institute, 3521 AL Utrecht, The Netherlands; 3UMCU-Wilhelmina Children’s Hospital—Child Health, 3584 EA Utrecht, The Netherlands; 4Center for Molecular Medicine, UMC Utrecht and Utrecht University, 3584 CX Utrecht, The Netherlands

**Keywords:** pediatric cancer, Wilms tumor, structural variation, chromosomal alterations, 1q gain, WGS, RNA-seq, cancer genomics

## Abstract

**Simple Summary:**

Chromosomal alterations and other structural variants have been recurrently identified in Wilms tumors (WT) and are promising biomarkers for risk stratification. Chromosome 1q gain occurs in one in three WTs and is associated with poor prognosis, but its impact on tumor biology remains unknown. Here, we investigated the mutational mechanisms and functional effects of chromosomal alterations in WTs, and in particular 1q gain. We identified subgroups of tumors with typical activated biological processes: muscle differentiation, immune system, kidney development and proliferation. Combining these subgroups with genomic data showed that tumors with 1q gain occur in all subgroups and can be associated with different functional effects. Also, 1q gain tumors differ in mutational mechanisms and co-occurring tumor-specific mutations. In conclusion, we identified subgroups of tumors with 1q gain and therefore propose that incorporating expression data in risk stratification could improve the clinical utility of 1q gain.

**Abstract:**

Chromosomal alterations have recurrently been identified in Wilms tumors (WTs) and some are associated with poor prognosis. Gain of 1q (1q+) is of special interest given its high prevalence and is currently actively studied for its prognostic value. However, the underlying mutational mechanisms and functional effects remain unknown. In a national unbiased cohort of 30 primary WTs, we integrated somatic SNVs, CNs and SVs with expression data and distinguished four clusters characterized by affected biological processes: muscle differentiation, immune system, kidney development and proliferation. Combined genome-wide CN and SV profiles showed that tumors profoundly differ in both their types of 1q+ and genomic stability and can be grouped into WTs with co-occurring 1p−/1q+, multiple chromosomal gains or CN neutral tumors. We identified 1q+ in eight tumors that differ in mutational mechanisms, subsequent rearrangements and genomic contexts. Moreover, 1q+ tumors were present in all four expression clusters reflecting activation of various biological processes, and individual tumors overexpress different genes on 1q. In conclusion, by integrating CNs, SVs and gene expression, we identified subgroups of 1q+ tumors reflecting differences in the functional effect of 1q gain, indicating that expression data is likely needed for further risk stratification of 1q+ WTs.

## 1. Introduction

Structural variants (SVs) and other large chromosomal alterations are often identified in pediatric cancer genomes but understanding their role in disease etiology is challenging. Gain of chromosome 1q (1q+) is of particular interest given its prevalence in embryonal tumors and associations with poor prognosis [1]. Also in Wilms tumors (WTs), the most prevalent kidney tumor during childhood, 1q gain has been recurrently identified and suggested as biomarker for risk stratification [2,3,4]. WTs are thought to arise from disrupted embryonic kidney development and typically consist of three histological components (stromal, epithelial and blastemal) in varying proportions and degrees of differentiation [5,6]. In addition to this phenotypic heterogeneity, somatic mutations have been reported in more than 40 genes, as well as chromosomal gains, losses and loss-of-heterozygosity (LOH) in a subset of WTs. Recent sequencing efforts have identified mutations beyond the classical Wnt pathway activating mutations (*WT1, CTNNB1* and *WTX*), affecting genes involved in developmental processes such as miRNA processing genes (*DICER, DROSHA*), kidney development transcription factors (*SIX1/2*) and genes involved in epigenetic remodeling or histone modification (*BCOR*, *CREBBP*) [7]. However, most of these single nucleotide variants (SNVs) occur with a low prevalence (<10% [7]). WTs individually carry few SNVs and previous studies have reported that no known pathogenic variants could be identified in a subset of tumors [8,9]. As a result, it remains challenging to identify which alterations and biological processes are clinically relevant and contribute to tumorigenesis, suggesting a role for structural variants (SVs) and copy number alterations (CNs). 

Risk stratification of WTs is mostly based on a combination of clinical and histological features of tumor cells remaining after preoperative chemotherapy as per SIOP-RTSG guidelines [5,6]. Fully necrotic tumors are stratified as low risk and tumors with blastemal predominance or diffuse anaplasia as high-risk. All other histological subtypes (stromal, epithelial, mixed/triphasic and regressive) are regarded as intermediate risk. Since a large number of relapses occur outside of the currently defined high-risk group, using molecular markers to further stratify the largest, intermediate risk group for more intensive treatment would likely contribute to reducing the relapse rate [5,6]. At present, the only molecular marker for poor prognosis is combined 1p LOH and 16q LOH, which is used in COG guidelines (without preoperative chemotherapy) to stratify favorable histology WTs [2,10]. However, 1p/16q LOH has a poor sensitivity since it is identified in only ~4–5% of cases and ~10% of relapses [2,10].

Recurrent chromosomal alterations are promising biomarkers and can help to gain insight into tumor biology, but a deeper understanding of their functional effects and underlying mutational mechanisms is needed to improve their clinical utility. Gain of chromosome 1q is highly prevalent in primary WTs (27–45%) and relapsed tumors (up to 75%) and evidence is accumulating for its association with poor prognosis [2,3,4,11]. However, it has been difficult to establish its prognostic value and the role of individual genes has not yet been elucidated. First, 1q+ can both occur as an early event in tumor development or acquired later, requiring up to three samples to detect its subclonal presence [12]. Second, the genomic context seems to matter, for example Haruta et al. reported that 12+ is associated with favorable outcomes also in the presence of 1q+ [13], but Gadd et al. recently reported the acquisition of 12+ in WT relapses [11]. Finally, 1q+ and 1p− can be the result of a single event (e.g., isochromosome or translocation) making it difficult to assess the prognostic value of 1q+ independently from 1p/16q LOH. These examples highlight that 1q+ tumors are a heterogeneous group, limiting the prognostic value of 1q+ without further exploration of its genomic context and functional effects.

Here, we investigated the genomic landscape and functional effects of somatic CN alterations and SVs in Wilms tumor patients using WGS data integrated with RNA-seq data. By focusing on primary diagnostic samples of intermediate and high-risk tumors we identified subgroups within the 1q+ patient population. Using genome-wide CN profiles, we identified three groups of tumors with different degrees of genomic instability: tumors with co-occurring 1p−/1q+, with multiple chromosomal gains and CN neutral tumors. Orthogonal to this, gene expression data grouped tumors with different somatic alterations in four clusters indicating similar functional effects in terms of biological processes affected. Intermediate-risk tumors split into three clusters: muscle differentiation, immune system, and early kidney development whilst all high-risk tumors clustered together. Tumors with 1q+ were present in all gene expression clusters demonstrating the differences in the functional effect of 1q gain. Furthermore, we identified distinctly different 1q+ mutational mechanisms and subsequent rearrangements among individual tumors. Our results show that combined analysis of CNs, SVs and gene expression data can identify subgroups of 1q+ tumors that converge on biological processes. This may therefore be promising for risk stratification.

## 2. Materials and Methods

### 2.1. Cohort Selection and Sequencing

Cohort selection was based on patients diagnosed with Wilms tumor for whom samples, subject to informed consent, were included in the Máxima biobank [14]. Tumors were classified by a pediatric pathologist into histological subtypes and risk groups based on viable tumor tissue after preoperative chemotherapy, as per SIOP-RTSG guidelines [5,6]. Patients were eligible when whole-genome sequencing (WGS) data was available from matching tumor-normal samples of sufficient quality. Sequencing library preparation and data pre-processing, including alignment and quality control, was done via the institute’s standardized pipelines and guidelines as described before [14,15,16]. In summary, high quality WGS samples were selected requiring a minimum median coverage of 27× for normal samples and 80× for tumor samples. RNA sequencing (RNA-seq) data was included when obtained from the same biosource as the tumor WGS and had at least 30 million unique reads. In total, for 30 patients WGS data was available with a median coverage of 36× for the normal and 104× for the tumor samples of which 22 patients also had RNA-seq data representative of the tumor with a median of 96 million unique reads.

To check whether samples were representative for the tumor, the frozen tissue samples used for molecular analysis were subjected to histological verification by a pathologist subject to material availability. For 10 out of 11 tumors, the histology of the sample was found to be representative of the tumor. However, not for tumor M829AAB, this diffuse anaplastic tumor sample lacked anaplasia and consisted of mostly stromal tissue with rhabdomyomatous differentiation, which was also reflected in the expression data (outlier) and therefore this RNA was excluded. Likewise, the RNA data was not included from three patients (M108AAD, M735AAA, M699AAB) where samples were taken prior to chemotherapy treatment.

### 2.2. Variant Calling

Somatic SNVs/indels were called using Mutect2 from GATK 4.1 [17] and filtered to a set of high confidence variants with tumor allele fraction >0.1, located on chromosomes 1–22 and X and not in ENCODE Blacklist poor mappability/high complexity regions [18]. Furthermore, we excluded variants in dbSNP unless they were also present in COSMIC [19]. Pathogenicity was predicted by variant effect predictor (VEP) (version 104) [20]. Copy number (CN) alterations were identified with the GATK4 pipeline following their recommended best practices [15]. To assess the copy ratio and minor allele fraction levels of chromosome arms, cytobands and gene bodies, we used weighted averages of the CN segments across these genomic intervals. Relative abundance of chromosomes was ascertained by comparing copy ratio and minor allele fraction of each chromosome arm relative to the maximum levels observed in the sample. 

Somatic SVs were called using Manta (version 1.6) [21], DELLY(version 0.8.1) [22] and GRIDSS (version 2.7.2) [23]. First, we selected SVs with at least seven supporting reads and removed those that had >90% reciprocal overlap with common (>1%) population variants retrieved from the NCBI repository (nstd166 [24], nstd186 [25]) and from DGV (version 2020-02-25) [26] accessed on 11 March 2021. Next, calls from the three tools were merged based on 50% reciprocal overlap and we required identification by at least two tools. To select variants that potentially had functional impact, we further filtered on allele fraction >0.1. Due to technical issues with running the tool, GRIDSS output was not available for two patients (M606AAA, M901AAC). 

### 2.3. Mutation Burden

The number of nonsynonymous somatic SNVs and indels per megabase (Mbp) was calculated to compare the burden of damaging small variants between tumors. Nonsynonymous SNVs/indels were included with VEP impact moderate or high in protein coding genes on chromosomes 1–22 (excl. chrX). Likewise, the number of possible nonsynonymous mutations (denominator) was calculated by summing all coding sequence bases in protein coding genes (41 Mbp). 

### 2.4. Mutational Signatures

The contribution of mutational signatures was assessed using the MutationalPatterns package (version 3.2.0) [27]. First, somatic SNVs were further filtered to prevent artifacts or products of selection. We required allelic depth >20 (DP >20) for both tumor and normal, removed variants with supporting reads in the normal, and removed all recurrent SNVs. Next, de novo signature extraction was performed on samples with >100 somatic SNVs (excl. M721AAA, M642AAA, M735AAA, M606AAA). As per recommended practices, we selected the number of signatures to extract based on the highest cophenetic correlation coefficient before it decreased (k = 3, cophenetic = 0.98) [27]. These signatures were compared against the known COSMIC signatures (version 3.2) [28] based on cosine similarity (>85%) and refitted to the complete cohort to obtain their relative contribution to the somatic SNV profiles of all tumors. 

### 2.5. Copy Number Clustering

Unsupervised analysis and clustering of copy number profiles was done with CNpare (version 0.99.0) [29]. Segmentation files were mapped into 1 Mbp genomic bins, excluding gvar/stalk/acen regions. Next, hierarchical clustering of samples was performed based on the Euclidean distance of these copy ratio profiles. The optimum number of clusters was derived from the elbow plot of within-cluster sum of squares (k = 3).

### 2.6. Expression Profile Extraction and Expression Clustering 

Non-negative matrix factorization (NMF, version 0.23) was used to derive expression profiles (or metagenes) that group genes sharing similar expression patterns throughout the cohort [30]. First, read counts were normalized with variance stabilizing transformation (vst) [31] after which the 10,000 most variable genes were selected, excluding genes on chromosomes X, Y and mitochondrial genes. Following recommended practices, the optimum number of profiles to extract was based on the highest cophenetic correlation coefficient before it decreased (k = 4, cophenetic = 0.98) [30]. Tumors were assigned to ‘expression clusters’ based on the highest relative contribution of these expression profiles. In addition, the expression clusters were verified with UMAP of the 10,000 most variable genes (Appendix A). 

### 2.7. Gene Set Enrichment of Expression Profiles

To compare the expression profiles in terms of the biological processes they reflect, we performed gene set enrichment analysis using CompareCluster [32]. Default settings were used to compare gene ontology terms of the top 1000 genes assigned to the expression profiles with the greatest log2-fold change. Significant terms were selected (q-value < 0.05) for each ontology (biological process, molecular function, and cellular component), relative to the 10,000 most variable genes corresponding to the gene set used as input for the expression clustering. For each expression profile, five representative ontology terms were selected to prevent overlap within each expression cluster. The number of genes of each expression profile used for ontology assessment is EX1 (780), EX2 (778), EX3 (729) and EX4 (610) because 24.5% of genes failed to map to Entrez identifiers.

### 2.8. Effects of Copy Number Alterations on Gene Expression

To assess the effect of copy number (CN) alterations on expression, we compared the expression values of tumors with a certain CN alteration against CN neutral tumors. We determined the copy number status of a gene by averaging the copy ratio values across the gene body and applying the default copy ratio threshold for inferring a gain or loss (+/− 0.2). We selected recurrently altered genes for which three or more tumors carried a CN alteration and assessed gain and loss events separately. The Wilcoxon rank sum test was used to assess whether the expression of CN altered genes was significantly different from CN neutral genes. VST-normalized counts were used as a measure of expression and *p*-values were adjusted for multiple testing with Benjamini and Hochberg (FDR < 0.2). We also required a minimum log2-fold change (l2fc) corresponding to 10% of the mean or median to select potentially biologically relevant changes (l2fc >+/− 0.138).

For individual tumors, we assessed whether a gene was affected by a CN/SV by comparing the tumor’s expression relative to those of CN neutral tumors. Hereto, a z-score of vst-normalized counts (nz-score) was calculated: *nz-score = (vst—vst_mean_neutral)/vst_sd_neutral.*

### 2.9. Cancer Genes

Cancer gene datasets were retrieved from COSMIC (cancer gene census v92) [19], OncoKB (accessed on 14 April 2021) [33] and Grobner et al. (2018) [34]. In addition, we obtained genes recurrently mutated in Wilms tumors from Table 1 from Treger et al. (2019) [7].

### 2.10. Wnt Signaling Pathway Analysis 

To assess Wnt signaling pathway activation, we considered the expression of the pathway genes as retrieved from MSigDB M39669 (version 7.4) [35,36]. To study relative differences among individual tumors, we defined the Wnt score. For this, we used the mean of vst-normalized counts of all Wnt signaling pathway genes, further scaled and normalized to a range of 0–1. 

### 2.11. Joint Analysis of Recurrent CNs/SVs and Recurrent Expression Changes

To investigate to what extent CNs/SVs contributed to the expression clusters, we analyzed whether increases in gene expression co-occurred with CN gains or nearby SV breakpoints within 1 Mbp. For example, the MYOG gene is assigned to profile EX1 and upregulated in EX1 tumors with a gain, therefore its upregulation could be CN/SV mediated. First, we selected genes upregulated in two or more tumors assigned to that expression cluster (nz-score >1.5, *n* = 2498 genes). We only considered upregulation because NMF extracts positive components (expression profiles). Next, we assessed whether these expression changes co-occurred with gains or nearby SVs in individual tumors (*n* = 581 genes) or in two or more tumors (*n* = 51 genes). Furthermore, we compared the set of CN/SV mediated expression changes with the full set of (expressed) genes with gains or nearby SVs (*n* = 272 genes) in two or more tumors. 

### 2.12. Gene-Level Integration of SNVs, CNs and SVs 

To assess whether a gene is affected by SNV/indels, CNs or SVs, we combined evidence acquired from these alteration types as reported in Appendix A and the oncoplots of Figure 1 and Figure 2. Each type of variant was first filtered to select the alterations that could affect gene function either directly (located in the gene body) or indirectly (nearby or spanning the gene). In case of possible indirect effects, we also required an expression change relative to the rest of the cohort (nz-score >+/−1.98). For SNV/indels, we removed variants predicted as benign by SIFT/PolyPhen unless they were also present in COSMIC (see Variant Calling). For SVs we considered breakpoints inside genes to have a direct impact, and required evidence from expression data for SVs breakpoints within 1 Mbp of the gene or SVs spanning the gene. In addition, CNs were regarded as indirect variants requiring expression change, and the copy ratio average over the gene body was used as its CN status. 

Finally, the alterations identified in a gene are summarized in the ‘alteration’ column: directly by SNV (snv) or SV bp (sv_bp), indirectly by spanning SV or SV bp within 1 Mbp together with change in expression (sv_indirect), or by CN gain or loss together with change in expression (gain or loss). These labels are simplified in ‘alteration simple’ where we distinguish only between snv, sv and cna. For example: downregulated gene with CN loss with SV breakpoints inside and nearby “sv_bp sv_indirect loss” and simplified as “sv cna”.

### 2.13. Visualization

Figures were generated using R and further edited with Adobe Illustrator. Packages used to generate figures: ggplot2 (version 3.3.5) [37], ggpubr (version 0.4.0) [38,39], pheatmap (version 1.0.12) [38], rstatix [40], circlize (version 0.4.13) [41].

## 3. Results

### 3.1. Copy Number Profiles Cluster Tumors into Three Groups Reflecting Different Degrees of Genomic Instability 

To study the genomic landscape of Wilms tumor, we analyzed WGS data of 30 paired tumor-normal kidney tissue samples (Appendix A). In this dataset we identified SVs, CNs, SNVs and indels. Gene expression was quantified using RNA-seq for a subset of 22 patients with matching DNA and RNA biosources, and excluding samples taken prior to preoperative chemotherapy (see Methods). Overall, WTs carry few mutations with a median of 6 somatic non-synonymous coding SNVs/indels (0.14/Mbp) and 5 SVs (Appendix A) and display genetic heterogeneity reflected in few recurrent alterations across cancer-related genes (Figure 1A). *CTNNB1, WT1* and *AMER1 (WTX)* are the most often recurrently mutated genes, in keeping with previous studies [7]. Most prevalent chromosomal alterations are gain of chromosome 1q (1q+), 11p loss-of-heterozygosity (LOH), 8+ and 12+ which occur in different genomic contexts (Figure 1B). WTs are not only genetically heterogeneous, but also several histopathological subgroups can be distinguished. 

Within the cohort, seven patients were stratified as high-risk and the remainder as intermediate risk (stage I–IV, *n* = 21 patients; stage V, *n* = 2 patients), according to SIOP-RTSG definitions [5,6]. The high-risk blastemal and diffuse anaplastic tumors have more unstable genomes than intermediate risk tumors, they carry more chromosome (arm) gains and losses (mean 3.9 vs. 1.2) and overall have a larger fraction of their genome altered by CN alterations (mean 15% vs. 5%) (Appendix A, Appendix A). SNV/indel burden correlates with age (*r = 0.55* Pearson correlation) and clock-like signature (SBS40, *r = 0.66*) (Appendix A). Furthermore, we identified platinum-associated signature (SBS31) in one tumor (M536AAA, stage V disease), and this sample was indeed the only one obtained after carboplatin treatment (Appendix A). In contrast, there was no correlation between SV burden and histological risk group or age. 

To investigate patterns of genomic instability, we clustered tumors based on their genome-wide CN profiles, resulting in three groups: tumors with co-occurring 1p−/1q+ (CN1), genomically unstable tumors carrying multiple chromosomal gains (CN2) and copy number neutral tumors (CN3) (Figure 1B,C). Comparing the CN clusters to histological classification shows that most epithelial and stromal WTs are copy number neutral and occur in CN3, which contains all tumors with mutations in both *CTNNB1* and *WT1.* Concurrent *CTNNB1/WT1* mutation was mutually exclusive to copy number gains and losses but does co-occur with 11p LOH. High-risk tumors tend to be more genomically unstable (CN1/CN2) as previously observed [42], although they occur in all three CN clusters. Integration with SV breakpoints suggests the CN clusters reflect differences in underlying mutational mechanisms (Figure 1B). Most full chromosome (arm) gains in CN2 tumors lack underlying SVs and likely result from mitotic errors, whereas segmental CN changes with nearby SV breakpoints were observed in both CN1 and CN2 tumors. In conclusion, we identified three CN patterns within the cohort and identified 1q gain either co-occurring with 1p loss (CN1) or multiple other chromosomal gains (CN2). 

Next, we investigated whether genes located in regions that are gained or lost in three or more tumors show changes in gene expression levels. Comparing gene expression of tumors with these recurrent CN alterations to CN neutral tumors resulted in a set of 756 differentially expressed genes (DEG) (>10% fold change, l2fc > +/− 0.138, false discovery rate (FDR) < 0.2) (Appendix A). This set also contains WT genes of interest such as *AMER1* (−0.76 l2fc, three tumors) and *WT1* (−0.87 l2fc, four tumors) (Appendix A). Overall, only 9% of the recurrently gained and 10% of recurrently lost genes showed significant differences in their expression relative to CN neutral tumors. This uncoupling of gene expression and copy number has previously been related to additional regulatory mechanisms (e.g., epigenetic changes) or differences in underlying mutational mechanisms [43] and could reflect differences in functional consequences of CN alterations.

### 3.2. Different Somatic Alterations Converge on Four Gene Expression Clusters 

To relate the heterogeneity in genomic alterations to affected downstream processes and pathways, we explored patterns in gene expression as a proxy for functional effects. Unsupervised analysis of matched RNA-seq data resulted in the identification of four expression profiles (sets of genes with similar expression patterns) (Figure 2A) and corresponding clusters of tumors (Appendix A). Each expression profile shows enrichment for distinct biological processes (q-value < 0.05) (Figure 2B, Appendix A). All high-risk tumors are in cluster EX4, which is characterized by high expression of genes involved in mitosis or proliferation. The other three expression clusters consist of intermediate risk tumors of various histological subtypes and are associated with muscle differentiation (EX1), immune system (EX2) or early kidney development (EX3). Next, we further characterized these expression clusters to investigate whether they can provide insights into the heterogeneous intermediate risk group. 

Biological processes reflecting aberrant kidney development are enriched in the expression profiles from both clusters EX1 and EX3. Cluster EX1 (*n* = 7 tumors) is characterized by increased expression of genes related to muscle cell differentiation, canonical Wnt signaling and mesenchymal stem cells and consists of tumors with stromal or mixed histology. In keeping with the expression pattern indicating Wnt pathway activation, this cluster contains all four tumors with co-occurring *CTNNB1/WT1* mutations as well as three tumors without these characteristic mutations that instead carry a 1q gain (Figure 2C). Furthermore, 11p LOH or loss is most prevalent in EX1 (five out of seven tumors) and absent in EX3 or EX4 tumors. Cluster EX3 (*n* = 7 tumors) is enriched for terms related to early kidney development, cilia organization and other genes related to epithelium or apical membrane. The only tumor with epithelial histology is present in EX3 together with mixed and regressive tumors. Of note, three tumors show significantly reduced *MIRLET7A* expression (M459AAA, M974AAC, M066AAB), suggestive of a differentiation block leading to preservation of the progenitor state [8]. Some EX3 tumors carry mutations in Wnt pathway genes, which suggests that Wnt pathway activation could be present in both EX1 and EX3 tumors albeit via different genetic alterations.

In contrast, clusters EX2 and EX4 do not show enrichment of the typical kidney developmental processes. Instead, cluster EX2 (*n* = 5 tumors) is characterized by the overexpression of genes related to immune system activation but also suppression, and signaling pathways including the *MAPK*, *ERK1* and *ERK2* cascade. Immune genes previously linked [44] to a suppressive tumor microenvironment in WTs (*TGFB1, IL10*) are highly expressed in all EX2 tumors whilst activating genes (*TNF, IL6*) show a more restricted expression pattern (Appendix A). EX2 consists of tumors carrying mutations in *MYCN* and *NONO*, that have been related to the preservation of the progenitor state [8], as well as epigenetic regulators/methylation (*BCOR, GNAS, KDM6A*) (Appendix A). Tumors in EX4 (*n* = 3 tumors) were stratified as high-risk based on histology and as expected, we identified *TP53* and *DROSHA* mutations in this group as well as deletions in 16q and 17p. Furthermore, we also identified *MYCN* mutations in EX4 tumors as well as alterations affecting other signaling pathways e.g., *JAK-STAT* and *Notch* (Appendix A). Taken together, the presence of an immunosuppressive expression signature might be characteristic of a relatively more high-risk subgroup within the current histological intermediate-risk group, but follow up analyses in a larger cohort are needed to further explore this relationship.

### 3.3. CNs/SVs Result in Wnt Pathway Activation

Activation of the Wnt signaling pathway is a well-established mechanism contributing to WT formation. Expression cluster EX1 is enriched for ‘canonical Wnt pathway signaling’, and indeed tends to show increased expression of downstream signaling molecules as well as Wnt antagonists relative to the rest of the cohort (Figure 3). To compare the Wnt signaling pathway activity in individual tumors, we used the mean expression of Wnt signaling pathway genes, scaled and normalized to a range of zero to one (Wnt score). As expected, the four *CTNNB1/WT1* mutated tumors of cluster EX1 show the strongest activation. Three tumors carry a hotspot gain-of-function *CTNNB1* mutation (p.S45F, p.S45P, p.T41A) and M035AAD carries p.H36P which is less common but also associated with nuclear accumulation of beta-catenin (protein product of *CTNNB1*) [45]. In each of these tumors, we identified multiple alterations in *WT1* indicative of disruption of both alleles irrespective of their disruption via either SNV or SV breakpoints. One tumor (M472AAC) carries a deleterious SNV whilst the other three tumors have SVs affecting *WT1*, often accompanied by loss of expression (Figure 2C, Appendix A). Although the underlying mutations are quite different among tumors, the similarity of their expression profiles suggests that they all have a disruptive effect on *WT1*. This highlights the importance of considering SVs as potentially pathogenic events, otherwise the WT1 disruption would have been missed in three out of four *CTNNB1/WT1* mutated tumors. 

Overall, we observed a range of Wnt pathway activation throughout the cohort reflected in coordinated upregulation of signaling genes (Figure 3, Wnt score). Tumors with hotspot mutations in *CTNNB1* and disrupted *WT1* have the strongest Wnt pathway activation, followed by a subset of 1q+ tumors in EX1 and EX3. However, these 1q+ tumors do not have a *CTNNB1* mutation or evidence of biallelic *WT1* disruption. Although we identified SNV/SVs in *AMER1*, these events alone are unlikely to be sufficient for Wnt pathway activation, especially when *AMER1* expression itself is unaffected (Figure 2C, Appendix A). We hypothesize that 1q gain could contribute to the Wnt pathway activation, for example via overexpression of *BCL9*, which is significantly higher expressed in 1q+ tumors and most pronounced in the EX3 1q+ tumors. Furthermore, in five of seven EX3 tumors, we identified alterations in genes previously associated with Wnt pathway activation (*SOX2, AXIN2, ZNRF3, BCL9, GPC3*) and moderate Wnt scores. In contrast, most tumors from EX2 and EX4 lack evidence of Wnt pathway activation. Taken together, in addition to the known concurrent SNVs in *CTNNB1/WT1*, we identified SNVs, CNs and SVs in EX1 and EX3 tumors that likely result in Wnt pathway activation.

### 3.4. 1q Gain Is Associated with Overexpression of Different Gene Sets 

Following our observations that CNs/SVs affect the same biological processes as known pathogenic SNVs, we investigated the contribution of CNs/SVs to the expression clusters. Hereto, we analyzed patterns of co-occurring increases in gene expression with underlying CN gains or SV breakpoints within 1 Mbp (Appendix A). For each expression cluster, we selected the genes from the expression profile that are recurrently upregulated in two or more tumors assigned to that cluster (nz-score >1.5, *n* = 2498 genes). Next, we focused on the genes recurrently altered by CN/SV in two or more tumors within a cluster (*n* = 272 genes). Combining these criteria, we identified 51 genes that are both recurrently upregulated and recurrently altered, representing 19% of genes with a recurrent CN/SV within the expression cluster (51/272, Appendix A). Vice versa, 2% of recurrently upregulated genes have a recurrent CN/SV (51/2498), likely due to low recurrence of CNs/SVs within the expression clusters. This agreed with our earlier observations that tumors group differently based on copy number data and expression data (Figure 2C), e.g., 8+ and 12+ are split over the expression clusters and 1q+ is present in all. When considering alterations in individual tumors, we identified underlying gains or nearby SVs in at least one tumor for 23% of recurrently upregulated genes (581/2498). Therefore, it seems that CNs/SVs contribute to the expression profiles, but their effects are not consistent and can differ among tumors. 

Gain of 1q is recurrent within EX1 and EX3 but associated with overexpression of different gene sets that reflect the biological processes characteristic for these clusters. We found that 1q gain is the underlying alteration for 48 of the 51 genes both recurrently altered and recurrently upregulated (Appendix A). Since these genes are in the expression profiles of either EX1 (*n* = 26 genes) or EX3 (*n* = 22 genes), this suggests that 1q gain can have different effects on gene expression (Figure 4A). In the 1q+ tumors of EX1, the recurrently upregulated genes are associated with GO terms “striated muscle cell differentiation” and “muscle contraction” (e.g., *MYOG, ACTA1, MYBPH, ACTN2, CACNA1S*). In contrast, 1q+ tumors of EX3 upregulate genes associated with “kidney development” and “epithelial cell differentiation” (e.g., *NPHS2, FLG, HRNR, REN*). Furthermore, these EX1 and EX3 genes are distributed across the chromosome arm, and we did not observe preferential localization of either gene set (Figure 4B). This demonstrates that 1q gain can affect different biological processes even if the same region is altered among tumors. 

### 3.5. 1q Gain Arises through Different Mutational Mechanisms 

Gain of chromosome 1q is the most recurrent chromosomal alteration in our cohort (27%) and of particular interest because of its suggested prognostic value [2]. Across all 1q+ tumors we identified 1q21.1–q32.3 (chr1:146,335,568–211,517,567) as a shared region but in different genomic contexts, such as co-occurring with 1p loss or with multiple chromosome (arm) gains (Figure 1). Using CN data, we inferred that 1q gain is clonally present in all 1q+ tumors (Appendix A). Furthermore, for tumors carrying multiple chromosome arm-level alterations with the same abundance, these were likely acquired through a single event. By combined analysis of CNs and SVs, we resolved profound differences in underlying mutational mechanisms or subsequent rearrangements of the eight 1q+ tumors (Figure 5A). 

First, we investigated the co-occurring 1p loss and 1q gain (five tumors), since previous studies suggested a mechanistic link [3]. Three tumors carry a stable full arm 1p−/1q+ which suggests they were acquired at the same time through isochromosome 1q (M536AAA, M067AAB, M975AAC). For the two tumors with focal 1p losses, we identified translocations and deletions that likely mediated these losses independently from their 1q gain (M459AAA, M974AAC, Appendix A). Consistent with these differences in mechanism, tumors with full 1p loss downregulate genes across the chromosome arm, whilst tumors with focal losses tend to downregulate genes nearby their SV breakpoints (Appendix A). These results show that the breakpoints of 1p−/1q+ CN gains and losses are important and that not all co-occurring 1p−/1q+ are the result of an isochromosome.

Next, we focused on structural variants affecting 1q (Figure 5A, Appendix A). In two tumors with partial 1q+, we identified underlying SVs that could have mediated the CN gain via a breakage-fusion-bridge cycle (M459AAA) and a large inverted duplication (M889AAA). Subsequent rearrangements were identified in two tumors, where we resolved translocations and large SVs that likely occurred after the isochromosome 1q or stable arm gain (M536AAA, M067AAB). In contrast, no large underlying or subsequent SVs were identified in the remaining 1q+ tumors (M974AAC, M975AAC, M754AAA, M010AAA). Our observations indicate that in all WTs analyzed the 1q+ is clonal regardless of the mutational mechanism resulting in 1q gain, and that diversification can take place due to subsequent SVs which is not visible from CN data alone. 

Finally, to complement the mechanistic differences, we set out to explore the effect of 1q gain on expression of cancer genes in more detail. Hereto, cancer genes on 1q were selected that are overexpressed in one or more 1q+ tumors relative to CN neutral tumors (nz-score > 1.98, Figure 5B and Appendix A, Appendix A). We focused on individual tumors given our previous findings that 1q gain was associated with increased expression of distinct gene sets in EX1 and EX3 tumors (Figure 4A). We identified cancer genes recurrently overexpressed across the cohort, e.g., *MDM4* in six and *CDC73* and *TPR* in five tumors, respectively (Figure 5B). The 1q+ tumors of EX3 share overexpression of seven cancer genes e.g., *BCL9* and *SETDB1,* whilst EX1 1q+ tumors share only *MDM4* overexpression and overexpress fewer cancer genes overall. Furthermore, some cancer genes located in the shared 1q gain region are overexpressed only in a single 1q+ tumor, which stresses the importance of considering gene expression profiles of individual tumors. For example, *ABL2* and *MCL1* in M754AAA, or *PARP1* in M974AAC. Taken together, we show that 1q gain has different functional effects through selective overexpression of genes. Further subclassification of 1q+ tumors is therefore crucial to elucidate the effect of 1q gain on tumor biology and this may redefine its prognostic potential. 

## 4. Discussion

Gain of chromosome 1q is one of the most prevalent chromosomal alterations in Wilms tumor and associated with poor prognosis in multiple pediatric cancers [1]. Gain of 1q is actively studied for its potential utility as a biomarker for risk stratification of WTs [2,3,4]. However, the underlying mutational mechanisms or functional effects of 1q gain on tumor biology remain unknown [11]. In this study, we integrated SNVs/indels, SVs and CNs with expression data to elucidate the biological and mutational processes active in primary Wilms tumors. Tumors profoundly differ in genomic stability and can be grouped into WTs with co-occurring 1p−/1q+ (CN1), multiple chromosomal gains (CN2) or CN neutral tumors (CN3). Gain of 1q was found in different genomic contexts (CN1 and CN2) and combined analysis with SVs showed that the underlying genomic rearrangement differed as well. Clustering based on expression data identified four groups of tumors that carry distinct genetic alterations but activate similar biological processes: muscle differentiation (EX1), immune system (EX2), early kidney development (EX3) and proliferation (EX4). Tumors with 1q+ are present in all four EX clusters and differ in which 1q genes are overexpressed. Recurrent 1q gains could be attributed to the overexpression of muscle-related genes in EX1 tumors and early kidney development genes in EX3 tumors. Furthermore, some tumors overexpress specific cancer genes located on 1q (e.g., *MDM4, ABL1*, *MCL1*) indicating that these patients might benefit from targeted therapies in the future [7]. These observations highlight the importance of precision oncology approaches implementing gene expression assessment because 1q gain does not always have the same effect. In conclusion, we showed that integration of WGS and RNA-seq reveals mutational and functional heterogeneity of 1q gain, which could indicate differences in affected biological processes. 

Wilms tumors are characterized by mutations that affect different biological processes, including Wnt pathway activation, miRNA processing, transcription elongation and epigenetic regulation [7,8]. Previously, Gadd et al. identified clusters of WTs related to disrupted kidney developmental processes and distinguished between the preservation of the progenitor state or abnormal induction [8]. They identified upregulation of 1q genes as one of the characteristics associated with the preservation of the progenitor state, but did not further specify candidate genes. Later, they also showed that alterations associated with preserving the progenitor state increased in prevalence at relapse [11]. Our EX3 cluster also contains tumors where expression data supports preservation of the progenitor state as they have upregulation of early kidney development genes and 1q genes as well as *MIRLET7A* downregulation. In contrast, cluster EX1 shows evidence of abnormal induction (mesenchymal-to-epithelial transition) reflected in Wnt pathway activation, overexpression of muscle differentiation genes and stromal or mixed histology. A subset of EX1 tumors carry *CTNNB1* hotspot mutations together with different *WT1* alterations, showing that disruption of *WT1* through either SNVs, SVs, or losses can have a similar effect. We show that SVs in *WT1* can be a potential explanation for tumors that lack SNVs in this gene but behave similarly to *WT1*-mutated tumors [46]. Three tumors in EX1 have no aberrations in *CTNNB1/WT1* and yet show Wnt pathway activation, which could be due to overexpression of *BCL9* and/or *Wnt9b* resulting from their 1q gain [47]. This association between *CTNNB1/WT1* mutations and muscle differentiation has been observed previously and suggested to be induced by chemotherapy or reflect developmental arrest in the mesenchymal stem cell-like precursor lineage [48,49]. Notably, the transcription factor *MYOG* located on 1q can induce terminal differentiation upon overexpression [50] and is highly expressed in both the 1q+ and *CTNNB1/WT1*-mutated tumors of cluster EX1. Taken together, we found that a subset of 1q+ tumors cluster together in EX1 with *CTNNB1/WT1*-mutated tumors based on gene expression patterns, which could indicate that they also have a similar cell of origin or susceptibility to chemotherapy. Therefore, we hypothesize that these EX1 1q+ tumors displaying muscle differentiation could represent a relatively lower risk subset compared to the 1q+ tumors in EX2–EX4 associated with preservation of the progenitor state, immune suppression or that resemble the high-risk tumors. 

The 1q+ tumors in our cohort share a commonly altered region (1q21.1–1q32.3) but integration of CNs and SVs revealed different mutational mechanisms underlying the 1q gain. We observed both tumors with SVs resulting in 1q+ and tumors without explanatory SV breakpoints that likely acquired 1q+ via mitotic errors, such as gain of the full 1q arm or isochromosome 1q (1p−/1q+). In our cohort, 1q+ was always clonally present and in the case of multiple clonal chromosomal gains, an early mitotic event such as multipolar division provides the most parsimonious explanation [51]. Simultaneous acquisition of multiple chromosomal gains followed by stabilization is also consistent with the observed uncoupling of gene expression with copy numbers, which is a known adaptation mechanism for tolerating aneuploidy [43]. Previous studies report 1q+ being both an early ancestral as well as a late subclonal event [12], or acquired during relapse [11]. Krepischi et al. identified a specific region of interest (1q21.1–1q23.2) which was associated with relapse [52]. Although the region overlaps, their candidate genes do not match our findings, apart from *CHD1L* overexpression in both EX3 1q+ tumors. These differences likely result from sample selection: they focused on the blastemal component and relapse samples whilst we characterized 1q+ tumors across histological subtypes and disease stages. Alternatively, late acquisition of 1q gain could have different functional effects that contribute to tumor progression and relapse.

Using SVs we explored suggested mechanistic links between 1q+ and 1p/16q LOH [3]. In three tumors, we identified 1p−/1q+ likely simultaneously arising via isochromosome 1q, and in contrast focal 1p deletions with underlying SVs in two tumors were likely independent from their 1q gain events. We did not identify SVs linking chromosomes 1 and 16. Furthermore, we identified subsequent rearrangements in a subset of tumors where SVs likely occurred after acquisition of 1q+ or 1p−/1q+ through mitotic errors. Although we observed co-localization of SV breakpoints and differentially expressed genes for the SV-mediated focal 1p losses, no such relationship was apparent for 1q gain (Appendix A). Nevertheless, these findings demonstrate the great diversity in alterations affecting 1q which has not been reported before. Further molecular profiling of 1q gains in a larger cohort using whole genome sequencing is required to fully understand the different 1q gain mechanisms and how the observed heterogeneity impacts gene expression.

Previous studies showed that 1q+ is associated with poor prognosis and more prevalent in relapse [11]. However, our results indicate that both the underlying mutational mechanisms and the biological processes affected by 1q gain can greatly vary among tumors. Therefore, the prognostic value of 1q+ is likely dependent on the genomic context and combination with other mutations [11] and there have been conflicting findings among studies regarding the role of specific chromosomes in relation to clinical outcomes [11,13]. Haruta et al. reported that 1q+ co-occurring with 12+ was associated with a favorable prognosis and overexpression of candidate genes [13]. We also analyzed two tumors with 1q+/12+ and they each overexpressed a distinct set of genes with only ~5% overlap. Both M754AAA (EX2) and M889AAA (EX1) overexpressed *MDM2*, but M889AAA also overexpressed four other candidate genes on 12+ associated with a higher overall survival rate (*BRAP, KDM5A, SFSWAP, ZCCHC8*) [13]. So, whereas EX1 tumor M889AAA might have a better prognosis, this seems less likely for M754AAA despite both carrying an extra copy of chromosome 12. On the other hand, Gadd et al. reported not only a high prevalence of 1q+ in relapsed favorable histology WTs (75%) but also of 12+ and combined 1q+/12+ compared to primary tumors. However, they did not confirm the previously suggested link between 1p−/1q+ nor report candidate genes [11]. These apparently contradictory findings suggest that 1q gain by itself might not be sufficiently specific as a marker for poor prognosis. Here, we also described substantial heterogeneity in mutational mechanisms, genomic context and gene expression effects among individual 1q+ tumors. Our results suggest that gene expression needs to be incorporated along with 1q+ status for improved risk stratification. 

## 5. Conclusions

Our investigation of copy number alterations and structural variants in Wilms tumors showed that diverse genetic alterations can converge on similar biological processes and specifically that 1q gain can have distinctly different effects on gene expression. Therefore, we propose the inclusion of gene expression data for further stratification of patients with 1q gain. While many studies include either only high-risk tumors or only favorable histology tumors, we characterized differences among mostly intermediate-risk tumors by analyzing primary samples after preoperative chemotherapy treatment. As our cohort consisted of recently collected diagnosis samples, follow up data to establish prognostic value are not mature as of yet. Given the low number of relapses, future results are needed from international studies with substantial cohorts, such as currently pursued by SIOP-RTSG and COG to determine the impact of 1q gain. In contrast to the first 1q gain studies utilizing targeted approaches to identify copy number gains [4], future studies should include expression data to distinguish subtypes of 1q+ tumors. Taken together, the potential prognostic value of recurrent CNs/SVs including 1q gain can be improved by characterizing events at the genomic level and integrating gene expression data. Genome-wide characterization of the full spectrum of alterations in WTs can help to acquire new insights into tumor biology and is essential for improving clinical utility of CNs/SVs.

## Figures and Tables

**Figure 1 cancers-14-04872-f001:**
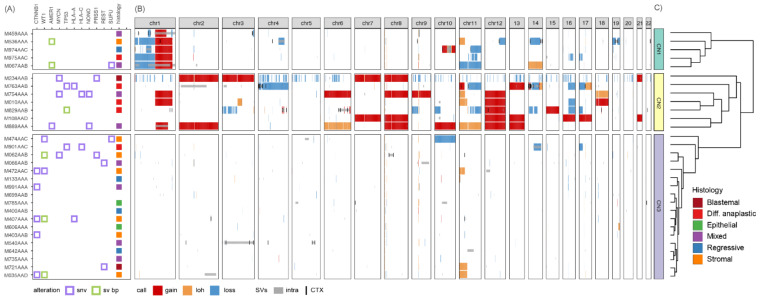
Copy number profiles cluster tumors into three groups reflecting different degrees of genomic instability. Unsupervised analysis of copy number (CN) data identified three clusters of tumors with distinct genome-wide CN patterns. Tumors that have co-occurring 1p loss and 1q gain (CN1), multiple chromosomal gains (CN2) or that are copy number neutral (CN3). (**A**) Oncoplot with cancer genes recurrently altered by SNVs (purple) or disrupted by SV breakpoints (green). Genes are ordered by the number of tumors they are mutated in. Tumors are annotated by their histological subtype. (**B**) Genome-wide CN profiles with gains (red), losses (blue) and copy number neutral loss-of-heterozygosity (gold), overlain by SVs with translocations (black lines) and intrachromosomal variants (gray). (**C**) Dendrogram resulting from hierarchical clustering of the CN profiles.

**Figure 2 cancers-14-04872-f002:**
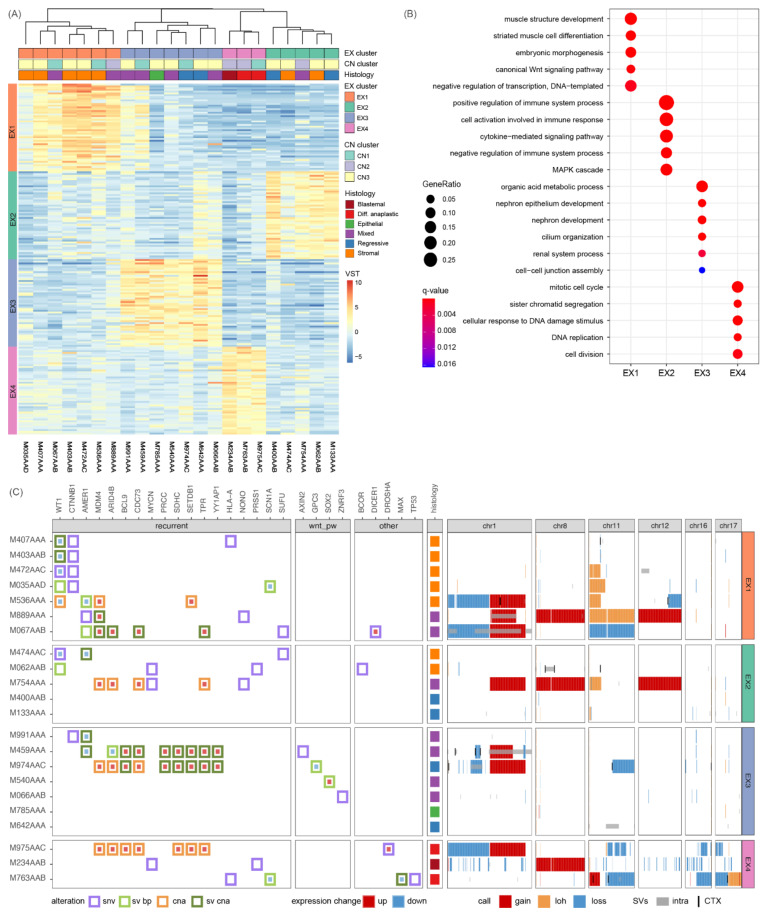
Gene expression data groups tumors with diverse genetic alterations that affect similar biological processes. Tumors were clustered in four groups (EX1–EX4) based on their resemblance to the four expression profiles identified by unsupervised analysis of the 10,000 most variably expressed genes (see methods). (**A**) Tumors (columns) of each expression cluster show upregulation of the genes (rows) of the corresponding expression profile compared to tumors of other expression clusters. Shown are the top 50 genes of each expression profile sorted by log2-fold change (l2fc). Tumors are annotated by their histological subtype. (**B**) Representative biological processes enriched in expression profiles (q-value < 0.05). (**C**) Oncoplot of tumors grouped by their expression cluster with alterations (left) and CN profiles with SVs (right), similar to Figure 1. The Oncoplot displays alterations affecting recurrently altered cancer genes, Wnt pathway genes and WT associated genes from Treger et. al. (2019) [7]. Alteration types: SNVs (purple) and SV breakpoints (light green), as well as nearby or overlapping SVs (dark green) and CNs (orange) in case of a gene expression change (nz-score >+/−1.98). The CNs and SVs are displayed for recurrently altered chromosomes 1, 8, 11, 12, 16 and 17, depicted as in Figure 1B. See Appendix A for gene alteration data and Appendix A and Appendix A for full sized figures.

**Figure 3 cancers-14-04872-f003:**
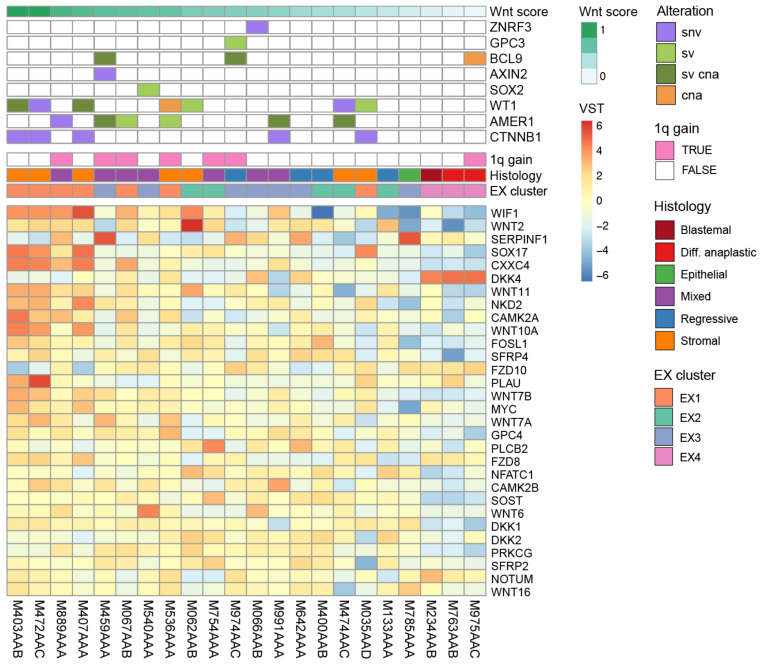
Wnt pathway activation resulting from distinct somatic alterations. Tumors (columns) display a gradient of Wnt pathway activation, ordered from left to right by the normalized mean expression of all Wnt pathway genes (Wnt score). Tumors are annotated with the alterations they carry (colors as in Figure 2C), and according to their 1q gain status, histological subtype and expression cluster membership. The genes (rows) displayed are the top 30 most variably expressed Wnt pathway genes across the cohort (MsigDB M39669).

**Figure 4 cancers-14-04872-f004:**
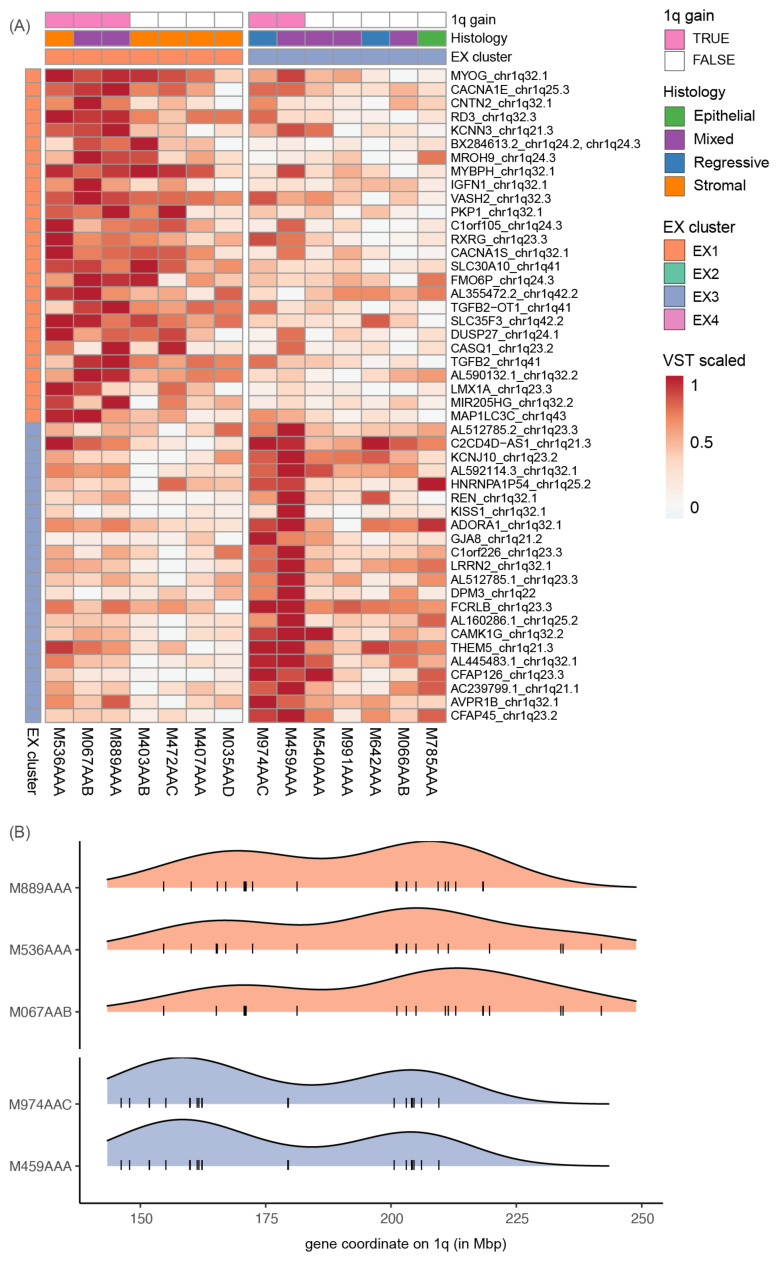
Wilms tumors with a 1q gain upregulate distinct gene sets. 1q+ tumors of EX1 and EX3 upregulate distinct gene sets corresponding to their expression cluster. Of the 51 genes recurrently upregulated (nz-score > 1.5) and recurrently altered by CNs/SVs within an expression cluster, 48 are located on 1q and assigned to either the EX1 (*n* = 26 genes) or EX3 (*n* = 22 genes) expression profiles. The EX2 and EX4 expression profiles did not contain genes located on 1q so therefore the tumors of EX2/EX4 are not shown. (**A**) The 1q+ tumors of EX1 upregulate genes assigned to expression profile EX1 (orange), and vice versa for 1q+ tumors of EX3 (blue). Expression values are scaled per row to highlight relative differences among tumors (columns) rather than between genes (rows). Tumors are annotated by their 1q gain status, histological subtype and expression cluster membership. (**B**) The 48 recurrently gained and upregulated genes (black lines) are distributed across the full 1q arm, and the density of these genes is similar for EX1 (orange) or EX3 (blue).

**Figure 5 cancers-14-04872-f005:**
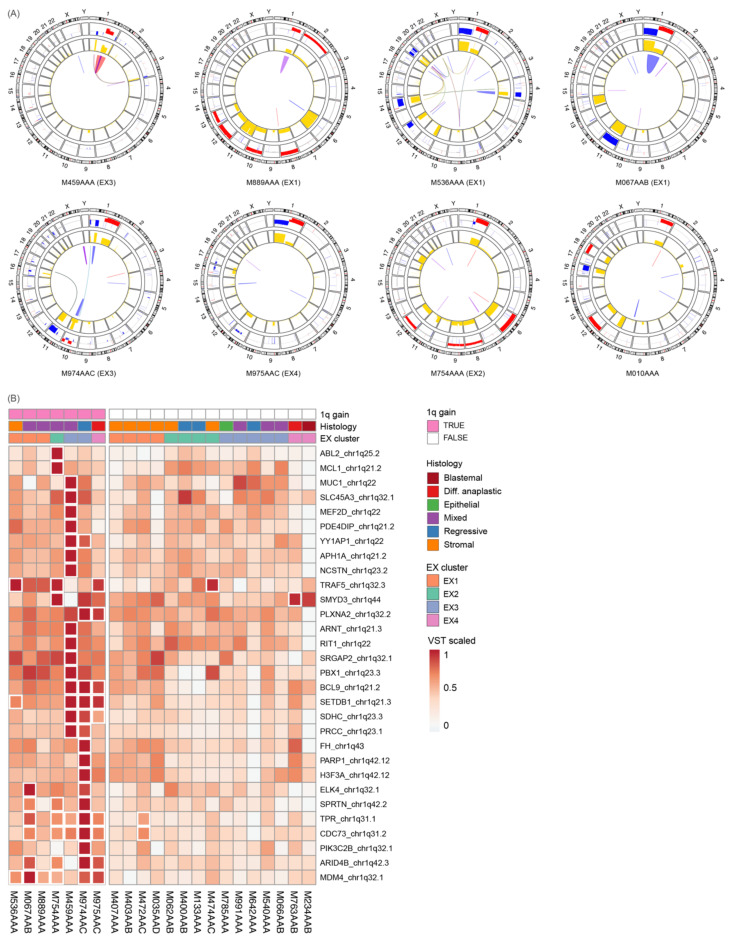
1q gain can arise through different mechanisms and result in overexpression of specific genes. (**A**) Tumors show different patterns in copy number data and structural variants affecting 1q. For two tumors, we identified underlying SVs: M459AAA has an amplification likely caused by a breakage-fusion-bridge cycle and M889AAA has a large, inverted duplication. The other six tumors carry a stable chromosome arm-level gain. For two of those (M536AAA and M067AAB) we identified SVs that likely occurred subsequently, since the SV breakpoints do not correspond to the CN segments. For the remaining four 1q+ tumors we did not identify large SVs or translocations. (**B**) All 1q+ tumors (pink) upregulate cancer genes located on 1q but also show large differences in which genes are significantly overexpressed (nz-score > 1.98, white border). Expression values are scaled per row to display relative differences among tumors rather than between genes. See Appendix A for locations of SVs and overexpressed genes on 1q.

## Data Availability

The WGS and RNA-seq datasets supporting the conclusions of this article are available in the EGA repository, dataset accession numbers EGAD00001009297 and EGAD00001009298. Code is available through https://github.com/princessmaximacenter/structuralvariation/ (tag v0.1), released on 12 August 2022.

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
