# Peer review of "Molecular Characterization Reveals Subclasses of 1q Gain in Intermediate Risk Wilms Tumors"

_cancers, 2022, doi:10.3390/cancers14194872_

Round 1

Reviewer 1 Report

The authors have described a highly detailed characterization of a small group of Wilms Tumours using WGS and RNAseq. My comments are mainly minor in nature. 

Firstly, the structure of the article should be improved - please consider using paragraphs to break up the introduction as well as parts of the results. Additionally, the final paragraph of the introduction contains a detailed summary of the results that could be removed. Likewise, the results section should be carefully read and repetition removed, likewise discussion sentences can be moved to the discussion. Figure captions are very long (e.g. Fig 2) and include results - can be shortened.

In the methods section, please describe or provide a reference for the Maxima biobank and make clear what has already been done and is carried out as part of this study (in terms of sequencing). This section is unclear.

Line 133 refers to "Cosmic" without reference, and later in the article "COSMIC" is mentioned - are they the same? Please unify and provide a reference. 

In Section 2.10, please explain what the "Wnt score" is - does it involve summing over all genes in the pathway, or an average value for them (for instance)? In Section 2.11 please be clear that the n's referred to are genes not tumours. Please also state the size of each of the 4 groups formed using the gene expression data in the text.

This study is observational not experimental, and the total number of tumours investigated is small (n=30 total), hence I find some of the statements that imply causality too strong. Are there public datasets that can be investigated to validate/confirm some of the findings? Potentially using array-based methods rather than sequencing? Additionally, the discussion mentions prognostic value of 1q+ - this was not investigated in the current study. Were those data available? It might be interesting, even though sample numbers are small. 

Author Response

The authors have described a highly detailed characterization of a small group of Wilms Tumours using WGS and RNAseq. My comments are mainly minor in nature.

Firstly, the structure of the article should be improved - please consider using paragraphs to break up the introduction as well as parts of the results. Additionally, the final paragraph of the introduction contains a detailed summary of the results that could be removed. Likewise, the results section should be carefully read and repetition removed, likewise discussion sentences can be moved to the discussion. Figure captions are very long (e.g. Fig 2) and include results - can be shortened.

> 1.1 We thank the reviewer for their suggestions and have revised the manuscript, including making the figure legends more concise and improving the paragraph structure.

In the methods section, please describe or provide a reference for the Maxima biobank and make clear what has already been done and is carried out as part of this study (in terms of sequencing). This section is unclear.

> 1.2 The Maxima biobank sequencing, alignment and quality control has been done with the institute's standardized pipelines and guidelines, which are elaborately described in our earlier manuscript (doi: 10.1101/2021.08.31.458342). We have updated the methods to reflect this. The analyses outlined in method section of this manuscript were carried out specifically for this study.

Line 133 refers to "Cosmic" without reference, and later in the article "COSMIC" is mentioned - are they the same? Please unify and provide a reference.

> 1.3 COSMIC reference has been added to the first mention in the text, and it is consistently referred to as "COSMIC". Reference for the COSMIC SBS signatures was also added (doi: 10.1038/s41586-020-1943-3)

In Section 2.10, please explain what the "Wnt score" is - does it involve summing over all genes in the pathway, or an average value for them (for instance)? In Section 2.11 please be clear that the n's referred to are genes not tumours. Please also state the size of each of the 4 groups formed using the gene expression data in the text.

> 1.4 The reviewer is correct, the Wnt score is the mean expression of all Wnt signalling pathway genes, scaled and normalized to a range of 0-1. Although this is detailed in the methods, we recognise it is important information to the reader and added it to the main text as well. We also included more details on the numbers of genes and tumors in the expression clustering analysis.

This study is observational not experimental, and the total number of tumours investigated is small (n=30 total), hence I find some of the statements that imply causality too strong. Are there public datasets that can be investigated to validate/confirm some of the findings? Potentially using array-based methods rather than sequencing? Additionally, the discussion mentions prognostic value of 1q+ - this was not investigated in the current study. Were those data available? It might be interesting, even though sample numbers are small.

> 1.5 We recognise the observational nature of our study and as such rephrased our findings and their implications in a more nuanced, hypothesis-generating way. In addition, we stressed the value of extensive follow up studies taking into account expression data as well as genomic alterations.

Although validation with an external dataset would be a valuable addition, to our knowledge no public dataset exists that allows for this comparison and contains the necessary gene expression data and 1q gain status of Wilms tumors. Furthermore, we regard the analysis of copy number alterations and structural variants as integral part of our study and believe that a follow up should also take this into account.

At present, survival analysis is not yet possible for this cohort as most samples have been collected over the past two years and the low relapse rate for this cancer type. Only one event has been recorded for this cohort: progression and death of M763AAB which was stratified as high-risk based on histology and did not have 1q gain. We added this information to Table 1.

Reviewer 2 Report

Very interesting and novelty study.

Author Response

Very interesting and novelty study.

> 2.1 We thank the reviewer for their comments.

Reviewer 3 Report

The team at Princess Maxima led by Dr. van Belzen, advanced this pilot study which posits that the high risk cytogenetic finding of 1q gain in Wilms tumors may be subdivided into biologically distinct subgroups, which they uniquely evaluated in a predominantly intermediate risk cohort, based on associated copy number alterations (i.e. in combination with 1p loss or multiple chromosomal gains) as well as broad transcriptomic changes which clustered preferentially with either muscle differentiation or early kidney development patterns of expression.  As both the COG and SIOP begin to implement risk stratification based on 1q gain, further delineation of which sorts of 1q gain are more/less high risk is prudent, or maybe amenable to particular therapeutic modalities is intriguing, and this effort certainly helps to advance our understanding of 1q gain in Wilms tumors.  The paper is well written and certainly thought provoking, and attempts to address a currently unmet need in the field.   The main limitation is the relatively small size of the cohort of patients with 1q gain, but this is the nature of a rare disease with a rare high risk subset being evaluated in subgroups.  My comments are as follows:

1. From Figure 1, it appears that 8 of 30 patients with WGS data and 7 of 22 patients with RNA-seq data actually have 1q gain.  Is that correct?  If so, can you please clarify why did Figure 4 only include 5 patients with 1q gain? 

2. On Lines 592-594, the authors note “Therefore, the EX1 1q+ tumors displaying muscle differentiation could represent a relatively lower risk subset compared to the 1q+ tumors in EX2-EX4”… this is a potentially very important point.  Since this is an internal cohort, are you able to provide clinical information on patient outcomes – specifically relapse or not?  Since 1q gain was not risk stratified in these cases, was there actually a different in EFS/OS?

3. Line 61: Please provide a reference for this statement, particularly the latter half: “WTs individually carry few SNVs while some have no known pathogenic variants at all”

4. Line 69 to 71: Combined LOH 1p and 16q is only clinically utilized as a high risk stratifier using the COG approach for newly diagnosed favorable histology Wilms tumors and, in that system at least on NWTS-5, combined LOH 1p/16q arises in only ~4-5% of cases (46 of 970 stage I/II; 30 of 686 stage III/IV in Grundy et al. J Clin Oncol, 2016) which I think actually supports your point more strongly that this is a rare event and, indeed, only used via the alternative COG system.

5. Line 88: Missing a space in several locations including Line 88 - “…effects.Here…”,  Line 282 – “…age.To…”, Line 311 “…instability.Unsupervised…”

6. Section 2.1 Cohort selection and sequencing: Consider moving some of these results into the results section

7. Line 224: Paragraph indentation missing

Author Response

The team at Princess Maxima led by Dr. van Belzen, advanced this pilot study which posits that the high risk cytogenetic finding of 1q gain in Wilms tumors may be subdivided into biologically distinct subgroups, which they uniquely evaluated in a predominantly intermediate risk cohort, based on associated copy number alterations (i.e. in combination with 1p loss or multiple chromosomal gains) as well as broad transcriptomic changes which clustered preferentially with either muscle differentiation or early kidney development patterns of expression.  As both the COG and SIOP begin to implement risk stratification based on 1q gain, further delineation of which sorts of 1q gain are more/less high risk is prudent, or maybe amenable to particular therapeutic modalities is intriguing, and this effort certainly helps to advance our understanding of 1q gain in Wilms tumors.  The paper is well written and certainly thought provoking, and attempts to address a currently unmet need in the field.   The main limitation is the relatively small size of the cohort of patients with 1q gain, but this is the nature of a rare disease with a rare high risk subset being evaluated in subgroups.  My comments are as follows:

From Figure 1, it appears that 8 of 30 patients with WGS data and 7 of 22 patients with RNA-seq data actually have 1q gain.  Is that correct?  If so, can you please clarify why did Figure 4 only include 5 patients with 1q gain?

> 3.1 The reviewer is correct. In Figure 4 we studied the association between upregulation of genes and underlying copy number/structural variants (CN/SVs) to see whether CN/SVs contribute to the expression cluster membership of tumors (EX1-EX4). The expression profiles for EX1 and EX3 contain genes located on 1q, so we looked at the tumors assigned to EX1/EX3 with 1q gain. This showed that the location of EX1 genes upregulated in EX1 tumors overlaps with the location of EX3 genes upregulated in EX3 tumors, as depicted in Figure 4. The two remaining tumors with 1q gain were not shown as they were assigned to EX2 and EX4 and the EX2/EX4 expression profiles do not contain genes located on 1q. We adjusted the legend of Figure 4 to explain this better.

Namely, "The EX2 and EX4 expression profiles do not contain genes located on 1q so therefore the tumors of EX2/EX4 are not shown."

On Lines 592-594, the authors note "Therefore, the EX1 1q+ tumors displaying muscle differentiation could represent a relatively lower risk subset compared to the 1q+ tumors in EX2-EX4" this is a potentially very important point.  Since this is an internal cohort, are you able to provide clinical information on patient outcomes ‚ specifically relapse or not?  Since 1q gain was not risk stratified in these cases, was there actually a different in EFS/OS?

> 3.2 We agree that survival analysis would be a valuable addition to our study. However this is currently not possible given the recent sample collection of our cohort and low relapse rate for this cancer type. (Also see response to reviewer #1 point 5)

Line 61: Please provide a reference for this statement, particularly the latter half: "WTs individually carry few SNVs while some have no known pathogenic variants at all"

> 3.3 The sentence has been rephrased to nuance this statement and a reference was added => "WTs individually carry few SNVs and previous studies reported that no known pathogenic variants could be identified in a subset of tumors (Gadd et al. 2017)."

Line 69 to 71: Combined LOH 1p and 16q is only clinically utilized as a high risk stratifier using the COG approach for newly diagnosed favorable histology Wilms tumors and, in that system at least on NWTS-5, combined LOH 1p/16q arises in only ~4-5% of cases (46 of 970 stage I/II; 30 of 686 stage III/IV in Grundy et al. J Clin Oncol, 2016) which I think actually supports your point more strongly that this is a rare event and, indeed, only used via the alternative COG system.

> 3.4 Thank you for providing these details and the reference. We have added these details to the text and refer to the papers from Gratias et al. J. Clin Oncol, 2016 and Grundy et al. J. Clin Oncol, 2005 which we believe are the references the reviewer is referring to.

"At present, the only molecular marker for poor prognosis is combined 1p LOH and 16q LOH, which is used in COG guidelines (without preoperative chemotherapy) to stratify favorable histology WTs[2,10]. However, 1p/16q LOH has a poor sensitivity since it is identified in only ~4-5% of cases and ~10% of relapses[2,10]."

Line 88: Missing a space in several locations including Line 88 – "effects.Here",  Line 282 "age.To", Line 311 "instability.Unsupervised"

> 3.5 We apologize for the inconvenience, these should have been new paragraphs. We have corrected the line break mistakes.

Section 2.1 Cohort selection and sequencing: Consider moving some of these results into the results section

> 3.6 We have updated the results section to now contain more information about sample selection. "Gene expression was quantified using RNA-seq for a subset of 22 patients for which samples were available matching the tumor DNA biosource, and excluding samples taken prior to preoperative chemotherapy (see Methods)."

Line 224: Paragraph indentation missing

> 3.7 Paragraph indentation and line break mistakes have been resolved.